# Peer review of "Phonon thermal Hall as a lattice Aharonov-Bohm effect"

_SciPost Physics, doi:SciPost Phys. Core 8, 061 (2025)_

## Round 1 · Author Response

The paper starts by an observation: Available experimental data indicate:
i) The peak temperature of the thermal Hall angle is concomitant with the peak temperature of the longitudinal thermal conductivity.
ii) This angle never exceeds an upper boundary.

It proposes that an explanation for both these features is to be found beyond the adiabatic and the harmonic approximations.
Feature (i) indicates that Normal phonon-phonon collisions are playing a key role.

As for feature number (ii), here is the chain of arguments:
a) Non-adiabaticity: Whenever the center-of-mass of the nuclei and the electrons do not coincide, the Born-Oppenheimer approximated wavefunction acquires a phase in the presence of magnetic field.
b) Anharmonicity: When collective atomic vibrations are anharmonic, this phase does not cancel out during periodic alternation between compression and expansion (or opposite transverse atomic displacements).
c) Quantitative assessment: The thermal Hall angle has an upper boundary set by the product of the phonon wavelength and the maximum atomic displacement. This matches the order of magnitude of the experimentally observed signal.

The central equation of the paper is its equation 16 , in which the phase of an acoustic phonon is linked to the phonon wavelength, the maximum atomic displacement and a dimensionless parameter q_e. The assumption that q_e ~1 can be realistic in common solids have been [strongly] contested by referee 1 and [less strongly] by referee 2. However, as I argue in the latest version of the paper, anharmonicity is sufficiently large in silicon to justify this assumption.

Below are replies to the latest round of review.

REPLY TO REFEREE 2:

I thank the referee for the time devoted to reading the manuscript and the constructive criticism. Indeed, the justification for Equation 16 in the previous manuscript was not clear. In the new version chapter 5 has been re-written. The field-induced phase of the BO approximated wavefunction of atoms arises when the center-of-mass for electrons and nuclei do not coincide. This can lead to a phase for phonons if expansion and contraction do not cancel each other.

In the previous version, I had argued that Gruneisen parameter is of the order of unity in silicon, and this leads to q_e~1. The referee did not find this argument convincing. The complexity of the charge distribution in silicon and the fact that negative charge does not simply circle ions did not impress the referee either. In the new version, I invoke the amplitudes of the Third Order Elastic Constants (TOEC) and the Second Order Elastic Constants (SOEC) in silicon. Combined with equations 21 and 22, they clearly demonstrate that atomic displacements (ether longitudinal or transverse) CANNOT be symmetric with respect to the equilibrium position.
I confess that I was not aware of the fact that in silicon the TOEC exceed by far the SOEC. However, this appears to be a common feature in solids. Surprisingly, it has not been discussed in scientific literature.

REPLY TO REFEREE 3:
I thank the referee for the time devoted to this manuscript and the positive evaluation.
i) Section 5 has been revised in depth to show how equations 17-20 lead to equation 16.
ii) The introduction new version contains the following disclaimer: “The focus of the present are insulators in which any spin–phonon coupling is absent and there is no magnetic contribution to the thermal Hall effect. This does not mean that such effects are absent in some of the materials included in Fig. 1b. Most probably, in presence of magnetism, other contributions to the thermal Hall conductivity may arise.”
iii) The three minor mistakes noticed by the referee, as well as a few others, have been corrected.

---

## Round 1 · List of Changes

-The abstract and section 5 have been entirely revised.
-A clarification on the focus of the paper (on non-magnetic solids) has been inserted in the introduction.
-Minor typos have been corrected through the text.

---

## Editorial Decision

published